# Tolerant Epitypes of Elicited Holm Oak Somatic Embryos Could Be Revealed by Challenges in Dual Culture with *Phytophthora cinnamomi* Rands

**DOI:** 10.3390/plants12173056

**Published:** 2023-08-25

**Authors:** Mar Ruiz-Galea, Carolina Kremer, Eva Friero, Inmaculada Hernández

**Affiliations:** Department of Agroenvironmental Research, Instituto Madrileño de Investigación y Desarrollo Rural, Agrario y Alimentario (IMIDRA), Alcalá de Henares, 28805 Madrid, Spain; mariacarolina.kremer@madrid.org (C.K.); eva.friero@madrid.org (E.F.); inmaher.sanchez@madrid.org (I.H.)

**Keywords:** somatic embryogenesis, epigenetic memory changes, priming, dual culture, *Quercus ilex*, disease tolerance

## Abstract

Holm oaks (*Quercus ilex* L.) can suffer severe infection by the oomycete *Phytophthora cinnamomi* Rands; the production of more tolerant plants is, therefore, required. Embryo formation is a key period in the establishment of epigenetic memory. Somatic embryos from three holm oak genotypes were elicited, either over 3 days or 60 days, with methyl-jasmonate, salicylic acid (SA), β-aminobutyric acid (BABA), or benzothiadiazole (all at 50 μM and 100 μM), or 10% and 30% of a filtered oomycete extract (FILT10 and FILT30) to activate plant immune responses. The number of embryos produced and conversion rate under all conditions were recorded. Some elicited embryos were then exposed to *P. cinnamomi* in dual culture, and differential mycelial growth and the progression of necrosis were measured. The same was performed with the roots of germinated embryos. Within genotypes, significant differences were seen among the elicitation treatments in terms of both variables. Embryos and roots of 60-day BABA, SA, or FILT10 treatments inhibited mycelium growth. The 3-day BABA (either concentration) and 60-day FILT10 induced the greatest inhibition of necrosis. Mycelium and necrosis inhibition were compared with those of tolerant trees. Both inhibitions might be a defense response maintained after primed embryo germination, thus increasing the likelihood of tolerance to infection.

## 1. Introduction

Holm oak (*Quercus ilex* L.), an evergreen oak, is distributed from Portugal eastward to Syria, and from Morocco and Algeria north toward France. It is an important forest tree and the main species associated with the *dehesa* or *montado*, agrosilvopastoral systems of southwestern Europe. Well-adapted to drought, this species is often the dominant type of vegetation in areas of transition between wet and dry climates [1]. However, longer droughts and the hotter temperatures now being experienced are having negative effects, including severe defoliation, attack by pathogens, and increased mortality [2,3], a scenario likely to become even worse as climate change advances.

The main pathogen associated with oak decline is the oomycete *Phytophthora cinnamomi* Rands [4,5,6]. This soil-borne microorganism causes root rot, bark cankers, the drying of twigs and branches, and ultimately death. The symptoms are similar to those caused by drought, high temperatures, or rotting of the root system. Nevertheless, the oomycete requires free water in the soil to sporulate and infect the plant. The disease tends to spread in a downward direction due to the movement of the causal agent with water [7].

Not all holm oaks are, however, equally affected. Some trees are more tolerant to *Phytophthora cinnamomi* and drought stresses, a consequence of genetic diversity between provenances and families within provenances [8,9]. The selection of those with the appropriate genetic background is necessary for use in traditional tree breeding programs [10,11]. Another way to address the increasing mortality problem is the vegetative propagation of tolerant plants to *P. cinnamomi* infection [12]. Micropropagation by somatic embryogenesis allows rejuvenated plants to be obtained with all the genetic potential of the originally selected trees. Furthermore, the clones produced can be used to study the responses of different genotypes to infection, and thus allow the most tolerant to be identified, along with the genes and proteins involved [13]. The genetic transformation of somatic embryos using a Gnk2-like protein may also allow genotypes with improved anti-oomycete activity to be obtained [14]. The induction of epigenetic changes on somatic embryos elicits a defense response and provides another opportunity to obtain tolerant holm oak trees [15,16]. There is increasing evidence that following exposure to an environmental stimulus, different genotypes acquire some kind of epigenetic memory that determines new behavior when restimulated. Stress exposure in plants leads to many epigenetic changes routed through mechanisms such as chromatin remodeling, DNA methylation, histone modification, and non-coding RNAs. Epigenetic markers are known to be transferred from plants to their offspring [17,18]. These epigenetic mechanisms can be used for developing resilient varieties [19].

Plants have many types of innate immune responses. These are often short-term, but infection may elicit acquired immunity by priming inducible defenses. This type of resistance is referred to as systemic acquired resistance (SAR), and plants are primed more quickly and effectively to adopt a defense response to the next pathogen attack [20]. Although some priming states disappear within a few days, others can even be transmitted to subsequent generations [21]. Indeed, transgenerational-induced resistance to *P. cinnamomi* occurs naturally in some holm oak trees [22] via maternally experienced stimuli at the time of seed formation, leading to epigenetic reprogramming [23,24]. The large number of genetically identical embryos that can be obtained by somatic embryogenesis [25] offers the opportunity to expose many initial explants to such priming. If the immune memory thus obtained can be maintained over development, primed seed could be produced for the production of pathogen-tolerant trees [26,27].

*P. cinnamomi* oomycetes secrete elicitins during the colonization process [28]. These proteins induce immediate defense responses in the host plant via hypersensitivity reactions. Cinnamomin, the elicitin used in this research, is one of the substances that have been proven to cause a defense response in oak plants [29,30]. Similar reactions may occur in response to other natural and chemically synthesized compounds, including β-aminobutyric acid (BABA), salicylic acid (SA), methyl-jasmonate (MeJA), and benzothiadiazole (or acibenzolar) (BTH), among others. The response induced depends on the species in question, its genotype, the type of elicitor used, its concentration, and the application time. The external application of these compounds may induce the accumulation of active oxygen species, phytoalexin biosynthesis, the activation of different defense-related enzymes, the production of phenolic compounds, or the appearance of pathogenesis-related proteins, some of which possess antimicrobial properties [31]. Primed plants do not lose their constitutive degree of defense but may gain greater resistance via the above treatment [18].

Different in vitro selection techniques for improving plant resistance are available for use with woody species [32,33,34,35]. Confronting host plant tissue with a fungal pathogen in dual culture has been used to assess the resistance of plant genotypes [36,37,38]. When the tissues of selected lines inhibit the growth of a mycelium, the explanation may lie in their production of externally exported resistance compounds. Dual culture assays have confirmed the chemotactic attraction of *P. cinnamomi* mycelia toward holm oak somatic embryos to be dependent on the latter’s genotype [15,16]. The same has also been observed in vitro with shoots and leaves [13] and in roots during the infection process in vivo [39]. The present work describes somatic embryo treatments by chemical elicitors or filtrates of *P. cinnamomi* oomycete to elicit a defense response that is maintained after embryo germination, increasing the likelihood of producing disease-tolerant trees.

## 2. Results

### 2.1. Multiplication Rate of Biomass and Production of Elicited Somatic Embryos

Three holm oak embryogenic lines (Q8, E2, and E00) were treated with elicitors that induce stress resistance, either 50 µM or 100 µM MeJA, SA, BABA, or BTH, or filtrates of a cinnamomin-inducing medium in which the oomycete had been cultured at dilutions of 10% or 30% (FILT10 and FILT30, respectively). For long-term elicitation, five vessels per genotype and elicitor were cultured on a semi-solid proliferation medium supplemented with each elicitor for 60 days. Short-term elicitation was applied with previous immersion in a liquid proliferation medium with elicitors for 3 days and then transferred to a semi-solid proliferation medium without elicitors for up to 60 days.

No significant differences were seen in the multiplication rate and the number of embryos produced between the elicitor concentrations of 50 and 100 μM for both short-term and long-term treatments. Therefore, only the values corresponding to 50 µM treatments are shown in Table 1. The number of embryos produced per vessel at the end of both treatments was dependent on the line genotype (*p* < 0.001) and less on the elicitor type (Supplementary Data, Table A1). Short-term treatment conditions reduced the embryo production of the E2 line (including controls), except for plant material elicited with MeJA. Long-term elicitation with 50 μM or 100 μM BTH resulted in necrotic embryos in all genotypes.

### 2.2. Dual Culture Assays with Elicited Somatic Embryos

Some elicited somatic embryos of all genotypes and treatments (except the short elicited E2 genotype, which was affected in embryo production) were challenged with the oomycete in dual culture. Treatment with the elicitors induced changes in the somatic embryos that were detected in terms of differential mycelium growth and the progression of necrosis compared with non-treated embryos.

The differential growth of the mycelium (DGM, i.e., growth toward the embryo, L, compared to growth away from it) (l) under each genotype/treatment type combination was recorded daily (Figure 1a).

The mycelium grew more strongly in the presence of somatic embryos than when embryos were not present (control). Differences between the genotypes were detected in the DGM values with unelicited somatic embryos (Table 2). Not enough E2 embryos were produced during short-term treatments (control and elicitors) to allow us to use this line in the DGM test shown in Table 2 and Figure 1.

With elicited embryos, there were also significantly different DGM values with respect to genotype (*p* < 0.001), elicitor (*p* < 0.001), and the elicitation period (*p* < 0.001). These differences were more evident on day 4 after dual culture initiation when the mycelium was reaching the embryos (Figure 2). The BABA treatments were associated with the smallest DGM values, particularly for short-term elicitation on the Q8 genotype (0.03 ± 0.06 cm) and long elicitation on E00 (0.27 ± 0.04 cm) and E2 (0.52 ± 0.14 cm). Differences in DGM with respect to the concentration of the elicitor used in long-term treatments could only be detected with Duncan’s post hoc test (Appendix A, Table A2).

The progression of necrosis in embryos was measured at 0, 24, and 48 h after mycelium contact with the embryo and according to a visual scale scored in Figure 1b. Values ranged from 4 for low necrosis (pale yellow embryos) to 10 for high necrosis (brown to black embryos).

The elicitation time affected the necrosis value at 0 h (*p* = 0.043), 24 h (*p* = 0.028), and 48 h (*p* < 0.001). Necrosis with long elicitation treatments was less than short-term elicitation at 0 h and 24 h but increased at 48 h (Appendix A. Table A3). Elicitors used significantly affected the necrosis (*p* < 0.001) at 48 h with differences in each genotype. Short-term elicitation with MeJa and FILT10 in genotype Q8 and BABA and SA in genotype E00 showed the lowest necrosis (Figure 3a). However, a long elicitation treatment with FILT10 reduced necrosis in embryos from all genotypes at 48 h compared to NT (Figure 3b).

### 2.3. Dual Culture Assays with Roots from Elicited Somatic Embryos

The genotype, elicitor, and treatment elicitation period affected the conversion (root and shoot formation) frequency of the elicited somatic embryos. Compared to long treatments, the application of short-term treatments reduced the conversion rates of NT and elicited embryos in E2 and E00 lines. Long elicitation with BTH also reduced conversion compared to all other elicitors (Appendix A, Table A4). The best conversion rate was obtained with long elicitation SA, BABA, and MeJA (Q8 conversion rates ranged from 31% with BABA to 45% with SA, E2 conversion rates ranged from 19% with MeJa to 32% with BABA, and the E00 conversion was 11% with BABA).

Further dual culture assays were performed using the roots of germinated somatic embryos that had been elicited with the aforementioned treatments (Figure 4). Due to the negative effect of some treatments on embryo production and plant conversion, we could not evaluate roots from the E00 and E2 genotypes subjected to short-term elicitation treatments, as well as the roots of any genotype subjected to long-term elicitation with BTH. Three Petri dishes per genotype and treatment with three root fragments were cultured on the opposite side of the assay plate from the mycelium. There were no significant differences in the growth of the mycelium and root necrosis between the elicitor concentrations of 50 and 100 μM for the same genotype and treatment. Therefore, only the values corresponding to 50 µM treatments are shown in Figure 4. There were significant differences in the growth of the mycelium (*p* < 0.05) among elicitation treatments at day 5 when the mycelium reached the roots. In the Q8 genotype, short-term elicitation with any elicitor produced DGM values lower than the control (Figure 4a). However, the lowest DGM values occurred with roots from a long elicitation with BABA, SA, and FILT10 (Figure 4b), in which embryos from Q8, E00, and E2 were pooled.

The extent of the necrosis of each root was also measured at 0 h, 24 h, and 48 h once the root had been reached by the mycelium. Significant differences were observed between elicitors at 24 h (*p* = 0.038) and 48 h (*p* = 0.003) after mycelium contact. Short-term elicitation with BABA, SA, and FILT30 (Figure 4c) or long-term elicitation with filtrates (Figure 4d) reduced necrosis at 24 h compared to the NT control and other elicitors. However, only long-term elicitation treatment with filtrates reduced necrosis at 24 and 48 h.

### 2.4. Dual Culture Results with Elicited and Naturally Tolerant Holm Oak Somatic Embryos

Embryogenic lines were established from holm oaks located in zones severely affected by *P. cinnamomi* in Plasencia, Extremadura (Spain). Somatic embryos obtained from naturally tolerant trees (T) in areas where disease was present and from control trees outside any focus of infection (P) were also grown in dual culture with *P. cinnamomi* mycelium. Figure 5 summarizes these results and, to facilitate the comparison, it includes the average values of the best elicitation treatments in terms of low DGM and necrosis, as well as the good production and conversion of embryos that were described so far in this manuscript. Daily DGM values with somatic embryos derived from tolerant trees and all elicitation treatments were lower than those recorded for the control (NT and P). Short-term elicited embryos had similar behavior to tolerant embryos. In addition, BABA and FILT10 inhibited mycelial growth on day 5 when approaching the embryos (Figure 5a). The necrosis was lower than NT in somatic embryos from the tolerant trees, embryos from 60-day FILT10, and embryos from all short-term elicitation treatments (Figure 5b). However, the tolerant trees and the short-term elicitation with SA or FILT10 in dual culture with the roots produced greater growth or attraction of mycelium than NT (Figure 5c). Root necrosis was also higher in tolerant trees and all elicitations, except in roots from 3-day BABA and 60-day FILT10 (Figure 5d). There was no total inhibition of growth mycelium or necrosis with elicited embryos or roots, but dual culture with *Phytophthora cinnamomi* Rand detected similar responses against the pathogen of tolerant trees and some elicitation treatments.

## 3. Discussion

When somatic embryos are forming by recurrent or cleavage embryogenesis under controlled conditions, they can enter a state suitable for introducing epigenetic markers. Kvaalen and Johnsen [40] first showed this in forest trees when they induced the formation of different epitypes of *Picea abies* by culturing embryogenic lines at different temperatures. Epigenetic changes in somatic embryos have been verified in other plant species, such as *Pinus radiate* and *Pinus pinaster* [41,42]. In the present study, the epitypes of holm oak could have been produced by the elicitation of somatic embryogenic lines. This was verified when somatic embryos of the same genotype behaved differently in dual culture with *P. cinnamomi* mycelium, depending on the elicitation treatment to which they had been subjected. Elicitation with 50 µM or 100 µM of BTH, BABA, MeJA, SA, or filtrate extracts at 10% or 30% (FILT10 or FILT30) for 3 or 60 days made the embryos show differential growth of the mycelium (DGM values) and the progression of necrosis with respect to untreated embryos. The genotype, elicitor, and elicitation time had an important effect on these results. A similar study with the same species could not find significant differences among elicitation treatments with filtered extracts and p-aminobenzoic acid (PABA), MeJA, and BTH at 5, 10, 25, and 50 μM [15]. However, in that report, only one holm oak genotype was used, and our results showed a high dependence on the genotype (*p* < 0.001) for DGM values.

Several elicitor-binding sites have been identified in cell plasma membranes and have investigated the general mechanism of action of elicitors [43]. Elicitors act as signal compounds at low concentrations, and it is important to know the application time for effective elicitation without possible toxicity or plant production alteration. No significant differences were seen in terms of biomass multiplication when using either the 50 or 100 μM concentrations of the chemical elicitors. However, the number of embryos produced was dependent on treatment and genotype. Embryos were necrotic after 60-day elicitation with 50 or 100 μM BTH. The same was previously reported when a concentration of 25 μM was used [15]. Long elicitation with BTH also prevented the germination of the few non-necrotic embryos produced. Embryo production of the genotype E2 was also particularly affected during the 3-day elicitation process, which may be due to stress caused by the agitation used [44], although it was not affected by treatment with MeJA. The exogenous application of MeJA improves growth and affects the levels of endogenous hormones, as well as the physiological and biochemical characteristics of stressed plants [15,45,46]. High levels of jasmonic acid (JA) can also stimulate the accumulation of indoleacetic acid and, therefore, the production of somatic embryos [47].

### 3.1. Differential Growth Measurements

The stimulation or inhibition of mycelium growth can allow the selection at the embryonic level of genotypes tolerant to infection [36]. In our dual assays, the mycelium grew toward both the embryos and their derived roots, suggesting it to be chemically attracted toward them. The DGM value was dependent on the genotype. It has been described that the constitutive levels of total phenols and condensed tannins may act as chemical defenses in holm oak, with variation between provenance and genotypes [48,49].

The development of dual cultures involving *P. cinnamomi* mycelium and elicited embryos or roots can provide the basis for the evaluation of tolerance of epitypes formed. Elicitation for tolerance was associated with lower DGM values and depended on the genotype and elicitation treatment but not the concentration of elicitor used. The recorded DGM values indicated that compared to the controls, the elicited somatic embryos inhibited the growth of the mycelium. With BABA treatments, significant differences in terms of less DGM were seen in all genotypes. BABA is a nonprotein amino acid that has been previously linked to priming mechanisms against *Phytophthora* sp. [50]. It is also known to induce defense responses in plants after treating their seeds. Certainly, treating tomato seedlings with BABA results in characteristic genome-wide changes in DNA methylation [51,52].

In dual culture assays involving the roots of elicited embryos, the growth of the mycelium was also reduced (lower DGM values) with respect to non-elicited controls. The present results confirm that the elicitation persists with embryo-derived roots. The elicitation of embryogenic lines with BABA also inhibited mycelial growth in root dual assays. The response to BABA is a popular model for studying the molecular signaling underpinning priming. This BABA-induced resistance (BABA-IR) is based on the priming of SA-dependent and independent defenses and is reported to provide broad protection against biotrophic and necrotrophic pathogens and abiotic stresses. So, it was confirmed to induce broad-spectrum resistance with epigenetic regulation and the long-term maintenance of priming [53,54,55].

Long treatment with salicylic acid (SA) or filtrate of a cinnamomin elicitin at 10% (FILT10) also reduced the growth of mycelium with roots. SA is a defense hormone required for both local and systemic acquired resistance in plants (SAR). The exogenous application of SA to induce resistance to pathogens through various pathways has been described [31,56]. Elicitins are also described as extracellular sterol-binding proteins with specific signaling responses in the host plant, and they have been widely used for in vitro selection of resistant plants [28,32,33,34]. Root exudates, like phytoanticipins, are produced prior to biotic stress against *Phytophthora cinnamomi* [57]. The inhibition of the mycelium could be justified by secreted compounds.

### 3.2. Progress of Necrosis

The correlation between the progress of necrosis and linear colonization has been suggested as a measure of tolerance. The root necrosis seen at 24 h and 48 h could easily reflect the degree of tolerance of plant tissues [58,59]. In our dual culture assays, long-term elicitation with FILT10 reduced necrosis in embryos at 48 h and in the roots at 24 h and 48 h of all genotypes compared to non-elicited controls. Some studies had already confirmed with quantitative real-time PCR a significant decrease in pathogen colonization of *Quercus sp.* roots after 24 h of pre-treatment with α- and β-cinnamomin [30]. In addition, great progress has been made in fungal elicitor-triggered plant immunity, especially in the signaling pathways of PTI (pathogen-associated molecular pattern-triggered immunity) and ETI (effector-triggered immunity) [60]. Short-term elicitation with BABA also reduced at 24 h and 48 h the progress of necrosis in embryos and roots. Although large differences were observed with respect to the genotype, the treatments coincided with the DGM inhibition results.

### 3.3. Tolerant Epitypes of Elicited Holm Oak Somatic Embryos Could Be Revealed by Challenges in Dual Culture with Phytophthora cinnamomi

The inhibition of the mycelium growth or the reduction in the progression of necrosis by tissues of the produced epitypes could reveal their tolerance to *P. cinnamomi* [36,37,38]. Tolerance is defined as the ability of the plant to mitigate the negative effects caused by the pathogen, despite an insignificant reduction in the presence and spread of the pathogen [57]. Somatic embryos from tolerant trees reduced mycelial growth and necrosis in dual culture respect population control. However, roots from tolerant trees attracted or increased mycelial growth, and they had high necrosis. Previously, other authors carried out a similar test with shoots and leaves from somatic embryos of tolerant holm oaks [13]. Shoots of one tolerant genotype but leaves of another tolerant genotype inhibited the growth of mycelium. Different responses were suggested by different phenolic content. Subsequently, results in dual culture with elicited embryos were compared with those in naturally tolerant trees. The short-term elicitated embryos with SA and FILT10 had the same behavior as tree-tolerant embryos.

Plant cells respond to *Phytophthora cinnamomi* infection by multiple defense actions. Within the same species and between genotypes, plant cells can display a slightly different defense response [57,61]. In dual cultures with pathogens, plant cells (embryos and roots) could also launch several signals during defense reactions. The first is the recognition of pathogen-associated molecular patterns by membrane receptors, leading to the activation of the innate immune response. Defenses could include antimicrobial compounds (less DGM) such as phenolic compounds, saponins, proteins and glucosinolates, and physical barriers to stop the progress of the pathogen within the host tissue (less necrosis). With roots, exudates may be the first line of defense against pathogens acting as inhibitors or attractant molecules (more DGM). Inhibitors were found in resistant but not moderately resistant avocado rootstocks [57]. To take roots fast enough was another strategy observed in tolerant genotypes [61]. If pathogens are recognized, plant cells can also be protected by SAR, with the activation of the hypersensitive response [62] characterized by local cell death (more necrosis).

Micropropagated clonal plants allow for minimizing differences in the analysis of plant defense [63]. Although it would be necessary to perform other advanced analyses to interpret the results, priming holm oak embryos for 3 and 60 days with 50 µM BABA or for 60 days with oomycete filtrates at 10% induced some tolerance (in a genotype-dependent fashion) to *P. cinnamomi* infection. Dual cultures could be the first simple procedure for testing holm oak tolerance. The inhibition of mycelium growth and necrosis can give useful information to select efficient elicitation treatments to produce disease-tolerant trees.

## 4. Materials and Methods

### 4.1. Somatic Embryos

Three holm oak genotypes were micropropagated for embryo elicitation: Q8 from Mora (Toledo, Spain) and E00 and E2 from the *Finca El Encín* in Alcalá de Henares (Madrid, Spain). Somatic embryogenesis was induced using teguments from developing ovules isolated from acorns [25]. For this, immature acorns were collected from adult trees in July, surface-sterilized with sodium hypochlorite (15%) and 1 drop of Tween^®^20 for 10 min, and then subjected to three washes in sterile water. At least 100 ovules/genotypes of 5–6 mm in length were excised and cultured in the dark at 23 °C on a proliferation medium without plant growth regulators. This medium contained the macronutrients of the Schenk and Hildebrandt medium [64], the micronutrients and vitamins of the Murashigue and Skoog medium [65], 30 g/L sucrose, and 0.6% agar (S1000, B&V, Parma, Italy). The pH was adjusted to 5.75 before agar addition, and the medium was autoclaved at 121 °C for 30 min. Developed zygotic embryos were extracted during the first month of cultivation. The teguments were then cultured on the same proliferation medium with monthly subculturing under a 16 h light photoperiod using a Sylvana GRO-LUX^®^ lighting system plus a Philips Cool-White system (120 μmol m^−2^ s^−1^). Induced somatic embryos were obtained in the third month of culture on the surface and cut areas of the teguments. Embryogenic lines were amplified by recurrent embryogenesis with monthly subcultures until elicitation assays could be started. Somatic embryos isolated from each culture were matured for 1 month on the proliferation medium containing 7 g/L of activated charcoal and then moved at 4 °C for 8 weeks in the dark. Embryos germinated on a fresh proliferation medium without plant growth regulators under a light photoperiod of 16 h at 23 °C.

Somatic embryos (without elicitation) were also obtained from selected holm oaks located in a severely affected area in Plasencia, Extremadura, Spain (39°58′01.6″ N–6°05′33.9″ W). Two asymptomatic genotypes were selected as tolerant trees (T), where the disease caused by *P. cinnamomi* produced high mortality in the rest [12] and other two genotypes were used as population controls (P) where *P. cinnamomi* was not present.

### 4.2. Elicitation Media

Somatic embryos were induced using different elicitors: methyl jasmonate (MeJA), salicylic acid (SA), β-aminobutyric acid (BABA), or benzothiadiazole (BTH) at a concentration of either 50 µM or 100 µM and an oomycete culture filtrate—a cinnamomin-inducing liquid medium—in which the strain of *P. cinnamomi* was cultured [29] and diluted to 10% or 30% (*v*/*v*) (FILT10 and FILT30, respectively).

For the oomycete culture filtrate, a virulent strain of *P. cinnamomi* (UEX1), donated by the University of Extremadura (Plasencia, Spain), was used. This had been isolated from a stand of diseased trees [66] and was maintained on a PDA agar medium (20 g/L Potato Dextrosa Agar and 5 g/L fresh agar) in darkness at 4 °C. Active cultures were obtained by taking 10 × 10 mm agar plugs containing mycelium and placing them in the center of Petri dishes containing a fresh PDA agar medium. These were cultured at 23 °C for one week in the dark. The oomycete filtrate treatments were based on those reported by Ganesan and Jayabalan (2006) [32]. First, an elicitin secretion medium (ESM) was prepared using 0.5 g/L KH_2_PO_4_, 0.25 g/L MgSO_4_.7H_2_O, 1 g/L asparagine, 1 mg/L thiamine, 0.5 g/L yeast extract, and 20 g/L glucose. This was sterilized by filtration through a 22 µm pore-size membranes. Then, 200 mL Erlenmeyer flasks containing 40 mL of ESM medium were inoculated with eight agar plugs (10 × 10 mm surface area) from active *P. cinnamomi* cultures on a semi-solid PDA medium. These were cultured under agitation (50 rpm) in darkness at 23 °C for five days before filtering the medium through a Büchner funnel. The filtrate was used the same day or stored at 4 °C (no more than 72 h). FILT10 and FILT30 were prepared by diluting accordingly with a proliferation medium.

### 4.3. Elicitation Treatments

Somatic embryos of each genotype were elicited via exposure to elicitors for either 60 days (long elicitation) or 3 days (short-term elicitation). Long elicitation involved culturing 0.3 g of embryogenic clumps for 60 days on a semi-solid proliferation medium supplemented with either 50 or 100 μM of each of the chemical elicitors (separately) or with the FILT10 or FILT30 preparations (replicated in quintuplicate). Incubation was then allowed at 23 °C with a photoperiod of 16 h of light. After the first month in culture, all the embryogenic material produced was cultured in new vessels, again using 0.3 g of tissue.

For short-term elicitation, 0.3 g of embryogenic clumps grown on a semi-solid proliferation medium were cultured for 3 days in a proliferation liquid medium supplemented with one of the elicitors. Five 200 mL Erlenmeyer flasks, containing 50 mL of the elicitation medium, were inoculated with embryogenic tissue, and the cultures were grown with agitation at 110 rpm for 3 days at 23 °C under a 16 h photoperiod. All embryogenic material from each flask was then filtered through nylon filters with a 40 µm pore size and transferred to a semi-solid proliferation medium without elicitors, which was then cultured for 60 days with one subculture at 30 days.

The monthly biomass multiplication rate (MR) was calculated as the number of new vessels needed to culture all the tissue produced by each treatment divided by the initial number of vessels (i.e., 5). The number of single somatic embryos produced per vessel (SE) was counted at the end of the 60-day period for each vessel, genotype and elicitor treatment, and means calculated. For both elicitation times, non-treatment (NT) controls were prepared in the same way, but without elicitors. Well-formed embryos were matured for 1 month and then were kept cold (4 °C) for 8 weeks, and they were then germinated according to a previously described protocol. The conversion rate (embryos with stem and root) was determined for each treatment.

### 4.4. Dual Culture Assays with P. cinnamomi

Elicited somatic embryos (about 500 mg) of each genotype and treatment were washed and placed on one side of a 90 mm Petri dish containing a 20 mL proliferation medium. A 10 × 10 mm plug of a PDA agar medium containing actively growing mycelium was placed on the opposite side, a quarter of the dish’s diameter from the respective edge. Assays involving somatic embryos of each genotype that did not undergo elicitation (NT) and cultures with only the mycelium were prepared as the controls. All tests were performed with 10 replicates. Once prepared, the assay plates were incubated at 23 °C in darkness for 10 days. Differential growth of the mycelium (DGM) was determined as the growth of the mycelium toward the somatic embryos (L) minus its growth in the opposite direction (l). Data were collected daily from day 2 to day 10. When the mycelium reached the embryos, the progress of necrosis was measured daily, according to a visual scale (Figure 3).

Similar assays were performed using three root tips (30 mm long, 2 mm wide) derived from the somatic embryos of the same genotypes (involving all elicitation treatments) that were allowed to germinate. The advance of the mycelium toward the roots was noted daily from day 2 to day 10, and DGM values were determined above. Lesion lengths were measured 24 h and 48 h after the mycelium reached each root (0 h).

### 4.5. Comparison of Elicited and Tolerant Genotypes When Challenged to Oomycete

Somatic embryos obtained from the naturally tolerant and control trees were grown in dual culture. DGM values on days 3, 4, and 5 with embryos and roots from holm oak somatic embryos were recorded in dual cultures with 20 repetitions per genotype. The progression of necrosis at 24 h and 48 h with embryos and roots was also recorded. Data were expressed as the mean value of 20 repetitions of tolerant genotypes, 20 repetitions of population genotype, and 6 repetitions for each genotype and elicitation treatment.

### 4.6. Statistics

Data recorded in the different experiments were subjected to analysis of variance using SPSS Statistica v.19 software (IBM Statistics). The normality was evaluated using the Shapiro–Wilk test, confirming the normal distribution of the data. The homogeneity of variance was assessed using the Levene test. According to the ANOVA results, when the F-ratio was significant, post hoc Duncan’s multiple range test analyses were performed to detect significant differences between genotypes and elicitors (*p* ≤ 0.05).

## 5. Conclusions

Tolerant epitypes of elicited holm oak somatic embryos can be revealed by the challenge in dual culture with *Phytophthora cinnamomi* mycelium. The elicitors induced changes in the somatic embryos that persisted in their derived roots. Treatments with β-aminobutyric acid (BABA) or long elicitation with filtrate of a cinnamomin-inducing medium, in which the oomycete had been cultured at dilutions of 10%, reduced the growth of the mycelium and the progress of necrosis in embryos and roots compared to other elicitors and the control. Epitypes of holm oak (*Quercus ilex* L.) tolerant to *Phytophthora cinnamomi* could be produced by the elicitation of somatic embryos.

## Figures and Tables

**Figure 1 plants-12-03056-f001:**
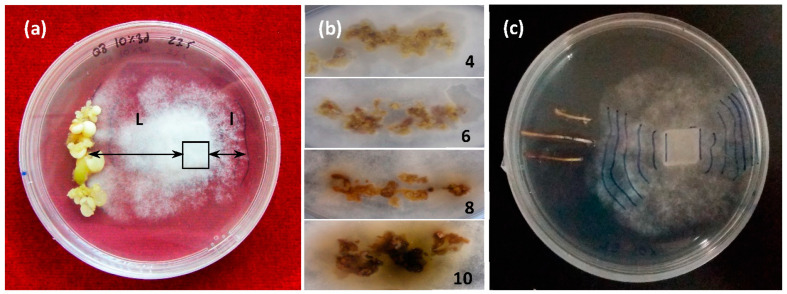
Challenges in dual culture with *Phytophthora cinnamomi* Rands; (**a**) dual culture involving holm oak somatic embryos and *P. cinnamomi* mycelium. Determination of the differential growth of the mycelium: DGM = (L − l) cm; (**b**) necrosis rating scale from 4 (low) to 10 (high); (**c**) dual culture of holm oak roots with *P. cinnamomi* mycelium and daily´s DGM marks.

**Figure 2 plants-12-03056-f002:**
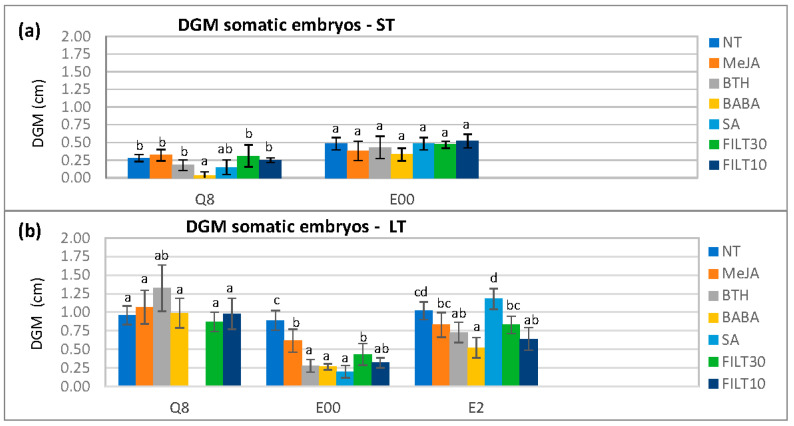
Differential growth of the mycelium (DGM) following the dual culture of holm oak somatic embryos treated with 50 μM of the different elicitors by short- and long-term treatments (ST and LT, respectively). DGM was evaluated on day 4 after dual culture initiation: (**a**) short elicitation in genotypes Q8 and E00 (the E2 genotype did not produce enough embryos for evaluation); (**b**) long elicitation in genotypes Q8, E00, and E2. NT, non-treated with elicitor; elicitors (50 μM.); MeJA, methyl-jasmonate; BTH, benzothiadiazole; BABA, β-aminobutyric acid; SA, salicylic acid; FILT10 and FILT30, filtrate of a cinnamomin-inducing medium at 10% or 30%. Data are means ± standard error of six repetitions. Bars with different letters are significantly different within each genotype using Duncan’s multiple range test (*p* ≤ 0.05).

**Figure 3 plants-12-03056-f003:**
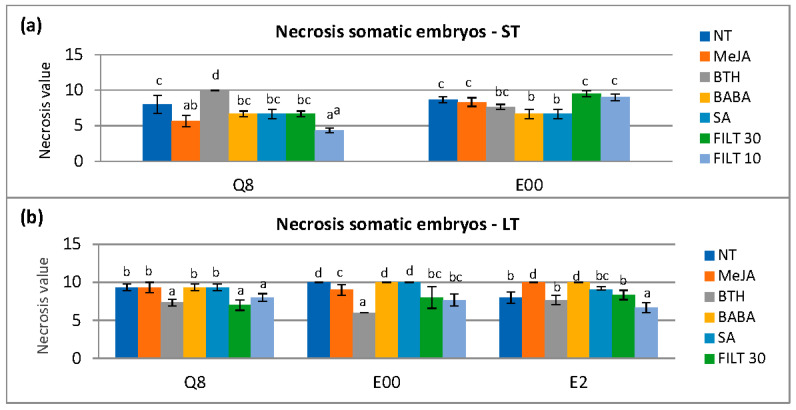
Necrosis in somatic embryos following dual culture of *P. cinnamomi* mycelium with holm oak somatic embryos treated with 50 μM of the different elicitors by short- and long-term treatments (ST and LT, respectively). Necrosis was evaluated 48 h after the *P. cinnamomi* mycelium contacted the embryos: (**a**) short-term elicitation in genotypes Q8 and E00 (the E2 genotype did not produce enough embryos for evaluation); (**b**) long elicitation in genotypes Q8, E00, and E2. NT, non-treated with elicitor; MeJA, methyl-jasmonate; BTH, benzothiadiazole; BABA, β-aminobutyric acid; SA, salicylic acid; FILT10 and FILT30, filtrate of a cinnamomin-inducing medium at 10% or 30%. Data are means ± se of six to ten repetitions. Bars with different letters are significantly different within each genotype using Duncan’s multiple range Test (*p* ≤ 0.05).

**Figure 4 plants-12-03056-f004:**
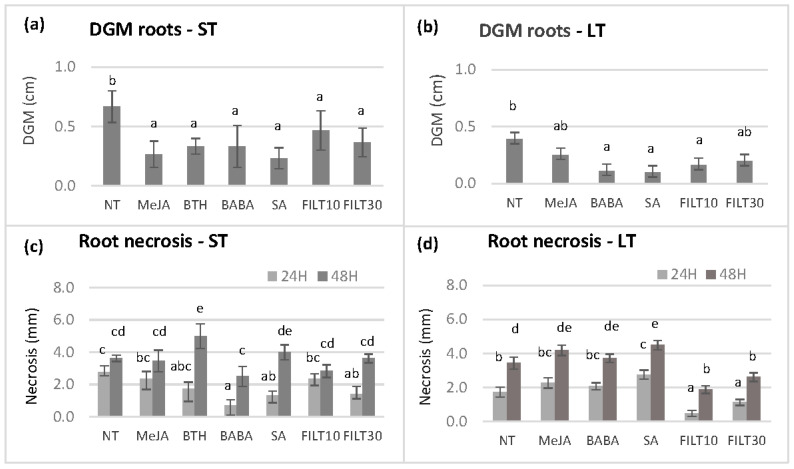
Differential growth of the mycelium (DGM) and root necrosis following dual culture of holm oak roots obtained from somatic embryos elicited by short-term and long-term treatments (ST and LT, respectively). DGM was evaluated at day 5 after dual culture initiation and root necrosis at 24 and 48 h after the mycelium had reached the roots (days 6 and 7 of dual culture): (**a**) DGM of roots from short-term elicited somatic embryos of the genotype Q8; (**b**) DGM of roots from long-term elicited somatic embryos of genotypes Q8, E00, and E2; (**c**) root necrosis observed on short-term elicited somatic embryos of the genotype Q8; (**d**) root necrosis observed on long-term elicited somatic embryos of genotypes Q8, E00, and E2. NT, non-treated with any elicitor; MeJA, methyl-jasmonate; BTH, benzothiadiazole; BABA, β-aminobutyric 210 acid; SA, salicylic acid (all at 50 μM); FILT10 and FILT30, filtrate of a cinnamomin-inducing medium at 10% or 30%. Data are means ± se of nine to twelve repetitions per genotype and treatment. Columns with different letters are significantly different between elicitors using Duncan’s multiple range test (*p* ≤ 0.05).

**Figure 5 plants-12-03056-f005:**
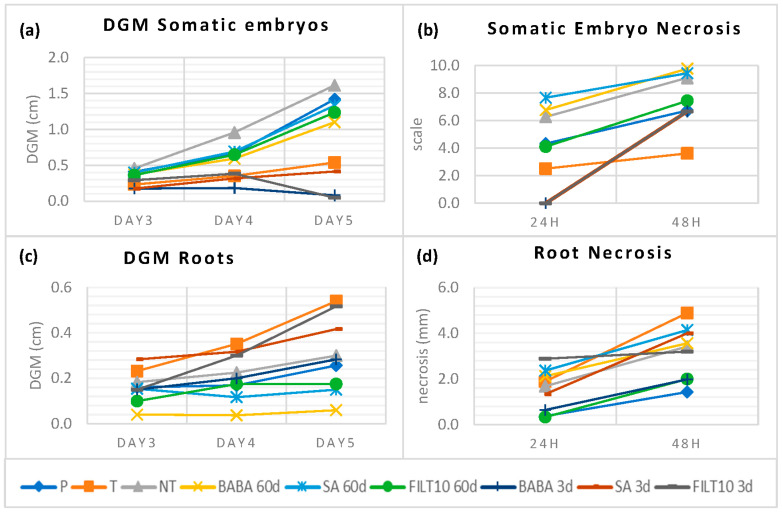
DGM and necrosis following dual culture of *P. cinnamomi* mycelium with elicited and naturally tolerant holm oak somatic embryos: (**a**) DGM using somatic embryos; (**b**) necrosis after mycelium contact with holm oak somatic embryos; (**c**) DGM using roots from somatic embryos; (**d**) necrosis after mycelium contact with holm oak somatic embryos. P: population trees; T: tolerant trees. NT: non-treated with elicitor. Short-term (3-day) and long-term (60-day) treatments with 50 μM: BABA, β-aminobutyric acid; SA, salicylic acid; FILT10, filtrate of a cinnamomin-inducing medium at 10%. In elicited genotypes, data are the mean of more than six replicates per treatment, and in naturally tolerant trees, there are twenty replicates per tree.

**Table 1 plants-12-03056-t001:** Effect of short-term elicitation and long-term elicitation on the multiplication rate (MR) and number of somatic embryos produced per vessel (SE) in Q8, E2, and E00 genotypes. NT, non-treated with elicitor; elicitors (50 μM); MeJA, methyl-jasmonate; BTH, benzothiadiazole; BABA, β-aminobutyric acid; SA, salicylic acid; FILT10 and FILT30, filtrate of a cinnamomin-inducing medium at 10% or 30%. Data are means ± se of five replications. Means within each column followed by different letters are significantly different according to Duncan’s multiple range test (*p* < 0.05).

Elicitor	Short-Term Elicitation (3 Days)						Long-Term Elicitation (60 Days)					
	Q8		E2		E00		Q8		E2		E00	
	MR	SE	MR	SE	MR	SE	MR	SE	MR	SE	MR	SE
NT	1.8	5.2 ± 1.4 a	1.6	0.0 ± 0.0 a	2.3	3.0 ± 0.3 a	2.4	7.7 ± 1.0 b	3.0	3.0 ± 0.8 b	1.8	3.7 ± 0.4 b
MeJA	1.5	3.5 ± 1.1 a	1.0	4.6 ± 1.4 b	1.5	3.6 ± 1.1 a	3.0	5.6 ± 1.0 b	2.4	6.0 ± 1.2 b	2.0	2.0 ± 0.6 ab
BTH	2.5	5.0 ± 1.9 a	0.2	0.0 ± 0.0 a	2.0	4.2 ± 2.5 a	1.8	0.4 ± 0.2 a	2.2	0.4 ± 0.2 a	1.0	0.6 ± 0.4 a
BABA	1.7	5.7 ± 0.6 a	1.6	0.4 ± 0.3 a	2.0	2.7 ± 1.8 a	2.8	6.5 ± 1.0 b	2.8	5.0 ± 1.2 b	2.4	2.3 ± 0.7 ab
SA	2.0	6.8 ± 2.3 a	0.8	1.0 ± 0.7 a	2.0	4.3 ± 1.0 a	2.4	2.9 ± 1.1 b	2.4	2.6 ± 0.8 b	0.8	3.3 ± 0.8 b
FILT30	1.7	5.9 ± 1.7 a	1.4	0.0 ± 0.0 a	1.5	3.4 ± 0.8 a	2.4	4.2 ± 0.9 b	1.2	3.3 ± 1.1 b	0.4	4.0 ± 0.0 b
FILT10	1.7	2.6 ± 0.7 a	1.6	0.0 ± 0.0 a	2.0	3.1 ± 1.0 a	1.6	4.7 ± 0.9 b	1.8	1.3 ± 0.7 ab	0.2	3.5 ± 0.5 b

**Table 2 plants-12-03056-t002:** Effect of genotype on daily differential growth of the mycelium (DGM) in dual culture with non-elicited somatic embryos from short-term and long-term treatments. Control: only the mycelium. Genotypes: Q8, E2, and E00, non-treated with elicitor. Data are means ± se of ten repetitions. Means within each column followed by different letters are significantly different according to Duncan’s multiple range test (*p* < 0.05).

Genotype	DGM (cm). Short-Term Treatment				DGM (cm). Long-Term Treatment				
	Day2	Day3	Day4	Day5	Day2	Day3	Day4	Day5	Day6
Control	0.0 ± 0.0 a	0.5 ± 0.3 a	0.5 ± 0.4 a	0.5 ± 0.8 a	0.2 ± 0.3 a	0.1 ± 0.4 a	0.1 ± 0.5 a	0.3 ± 0.5 a	0.7 ± 0.5 a
Q8	0.0 ± 1.0 a	2.5 ± 0.9 a	2.8 ± 0.5 b	0.7 ± 0.6 a	2.2 ± 0.4 b	4.4 ± 0.7 b	9.6 ± 1.3 b	17.2 ± 1.3 b	23.4 ± 1.4 b
E00	0.8 ± 0.5 a	3.7 ± 1.2 b	4.8 ± 0.9 c	2.0 ± 1.3 a	0.2 ± 0.5 a	4.3 ± 1.0 b	8.9 ± 1.3 b	14.4 ± 1.7 b	19.8 ± 1.3 c
E2	-	-	-	-	1.3 ± 0.2 b	5.0 ± 0.6 b	10.2± 1.2 b	16.8 ± 1.6 b	22.1± 0.5 bc

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
