# Peer review of "Tolerant Epitypes of Elicited Holm Oak Somatic Embryos Could Be Revealed by Challenges in Dual Culture with Phytophthora cinnamomi Rands"

_plants, 2023, doi:10.3390/plants12173056_

Round 1

Reviewer 1 Report

The ms includes valuable methodology and results that deserve to be published in Plants. Although many data related to elicitation are analyzed and presented in the manuscript, there are some problems that should be amended before being accepted in a journal such as “Plants”.

The first and important problem is the quality of English Lenguage that makes the ms sometimes illegible.

Other suggestions

Abstract

Line 19.  Add “somatic” embryos

Line 21. Explain what “similar to..” means

Introduction

Line 37 -Pay attention to references [4-7]

Results

Line 116. Table 1.  Filtrate “from”.

Table 1.  Use short-term and long-term.

Line 118. Table 1. Please include the number of replications. Data are means ± SE of xxx replications

Line 130.  Figure 3,  should be Figure 1?

Line 34 Table 2. It is not clear whether these somatic embryos were elicited or not. If it represents a short-term elicitation are you represented mean data? Please explain

- Figure 1. a. Tthe statistic after this Figure is confusing Why do authors present the mean data (dark blue bars)?.

-  Also, are authors presenting a full action among genotypes and elicitors?, then I do not understand mean separation

- Figure 1b. Data from E2 genotype are missing. Why the mean bar is added?

- Line 152.  The visual scale to measure necrosis should be explained (in addition to Figure 3b) otherwise, the method cannot be used by other researchers. Again mean bar is included, why?

- Root assays. These assays were not performed with the E2 genotype?. Please explain

- Table 4 a. mean separation was applied to the average among genotypes? Then, there is not a significant interaction between the genotype  and the elicitors). It seems that  E00 is not affected whereas Q8 does it.

- Line 214 to 234- Comparing results from elicited and tolerant genotypes is a very interesting point. But results should be presented in the same table or Figure. Authors should prepare Figures where roots from elicited somatic embryos are compared to those from tolerant genotypes before make conclusions.

Material and methods

- Line 373. What does “cofactors” mean?

- Line-385, please included more information about where in Extremadura is that a small area in Spain?

- Line 396-to 403. This paragraph should be removed or shortened.

- A section that included methodology to compare elicited vs. tolerant genotypes when challenged both to the oomycete should be included

The ms needs extensive English editing

Author Response

Sincerely

M. Ruiz Galea

Reviewer 2 Report

I find the job interesting enough. This article describes the production of somatic embryos from holm oak elicited with methyl jasmonate, salicylic acid, β-aminobutyric acid (BABA), or 10% or 30% dilutions of a filtrate of a culture medium in which Phytophthora cinnamomi (FILT10 and FILT30), to obtain seedlings tolerant to P. cinnamomi.

Line 108: change the abbreviation ES with SE.

Line 144: The authors report that the few values recorded for the E2 embryos were not used in short elicitation. Why are the data of E2 however reported in Table 2?

Line 1182-183: As reported in Appendix A Table 4, short elicitation reduced the production of embryos to zero for lines E2 and E00. How was it possible to produce roots from all three lines?

Line 493: for F and p-level values a 0 is missing in all the tables in Appendix A.

Line 551: in Table A4, arrange the columns in the following order: Q8, E2, and E00, in the same order as Table 1.

The analysis of variance (ANOVA) and Duncan’s test assumes that data are normally distributed (Gaussian distribution) and have equal variance/covariance. With what statical tests did the authors verify these assumptions? Please explain in the text. If the data can be analysed with parametric tests, I suggest replacing Duncan's test with Turkey's test. If not, analysis of variance should be done with Kruskall Wallis test, followed by Dunn's post-hoc test

Author Response

Sincerely

M Ruiz-Galea

Reviewer 3 Report

The manuscript presents a substantial amount of scintific study. However, experimental design lacks very critical points of validation. 

E.g - There were controls missing in the experiment design. It is important to conduct a validation experiment by inoculating the elicitor and fungi to see whether the chemical itself supress the growth or it is actually the elicited embryo supresses the fungi.

The authors claim that they have found tolerant epitypes through somatic embroyos with the treatment of elicitors, but the results are contradicting. There is no tolerance shown but all somtic embrayos were nectrotic with time.

Data presentation has some issues as well. Experimatal procedure can be more clear.  

Overall manuscript is written well, easy to follow and interesting to read. 

Author Response

Sincerely

M Ruiz Galea

Round 2

Reviewer 1 Report

Plants -2428191 -R1

The ms has been improved. I would suggest the following minor revision

Line 43.  “Some trees are more tolerant of the above stresses…”. The sentence has no sense. Please rewrite it.

Line 66. Please include “to” the next …

Table 1. Please use the same decimals  (2.0)…

Table 2. Plese use “.” Instead “,” for decimals

Figure 1. Line 158. Please note that “columns with different letters indicated…

The same for the other figures since “Different letters are significantly… has no sense”

Discussion

Line 249. Do authors mean  Epigenetic changes or other changes, please explain?

Line 259. reference number 14 is related to transformation experiments rather to elicitation.

Line 312. Root exudates like… are?

Line 321- 48h in

Line 361. Please use analysis

Material and methods

Line 380. Fe-EDTA are micronutrients

Line 381. Please explain de agar brand

Appendix  A Table A1.

Please explain what Factors are ( 1= elicitor concentration; 2 stands for genotype?...

The English language has been improved

Author Response

See attachment. Thank you again for your suggestions.

Sinceresly

M. Ruiz-Galea

Reviewer 2 Report

The author adequately answered  my revision and now the manuscript may be accepted in present form  

Author Response

See attachment. thank you again for your suggestions.

Sinceresly 

M. Ruiz.Galea
